# Quality Evaluation of Poultry Litter Biochar Produced at Different Pyrolysis Temperatures as a Sustainable Management Approach and Its Impact on Soil Carbon Mineralization

**Chen-Chi Tsai \*** and **Yu-Fang Chang**

Department of Forestry and Natural Resources, National Ilan University, Ilan 26047, Taiwan; yufang0115@gmail.com
\* Correspondence: cctsai@niu.edu.tw; Tel.: +886-3-931-7683

**Abstract:** Poultry litter biochar (PLB) is a value-adding soil amendment and an economically sustainable approach that is used to enhance food safety and reduce environmental harm. Poultry litter biochar has promising potential but has been under-examined in regards to carbon (C) sequestration in relation to its type and application rate. The objective of this study was to investigate the effectiveness of PLB in enhancing the C sequestration of acid soils through a short-term incubation experiment. The soil was amended with different materials: PLB (1%, 5%, and 10%) and a control (non-amended). The results indicated that PLB application increased soil C mineralization relative to the control (19–1562%), it significantly increased with an increasing application rate (e.g., increased addition 29, 99, and 172% for 1, 5, and 10% of 400 °C PLB), and the soil C mineralization and applied carbon mineralized (ACM) significantly decreased with temperature (e.g., the cumulative C pool ranges of ACM with 1% PLB, added at pyrolysis temperatures of 200, 300, 400, 500, and 600 °C, were 42.0, 34.4, 19.6, 6.16, and 4.04%, respectively). To assist sustainable soil management and to aid the achievement of multiple sustainable development goals (SDGs), as well as to maximize the benefits of PLB applications and minimize the potential environmental risk, it is suggested that application of PLB, pyrolyzed within 400–600 °C at a rate between 1% to 5%, should be adopted in acidic soils in Taiwan.

**Keywords:** poultry litter biochar; sustainable development goals (SDGs); sustainable soil management; C sequestration; soil amendment

## 1. Introduction

Poultry litter (PL), the mixture of poultry manure and bedding material from poultry farms containing high levels of organic carbon (OC), nitrogen (N), phosphorus (P), potassium (K), and other plant nutrients has been widely used by farmers, especially vegetable growers, as a source of plant nutrients and as a soil amendment [1–3]. In Taiwan, the annual generation of PL is 2.3 million tons, accounting for 32.3% of the country's total agricultural waste, and the majority is disposed of as an organic fertilizer through land applications in Taiwan and elsewhere. However, the improper management of PL in small acreage land areas, including repeated and excess application, are responsible for severe nutrient accumulation in soils and subsequent runoff and leaching losses in water bodies, which causes further food safety concerns and environmental harm [4–6]. Using PL as a pyrolysis feedstock has advantages over typically used, plant-derived materials because it can be cheaper and can alleviate the sustainability concerns related to using purpose-grown biomass in the process [4]. Many studies demonstrated that the conversion of PL to PLB is a farm-based, value-adding approach used to: recycle organic waste, slow down nutrient release, maintain a more constant and longer-term supply of nutrients in soil, prevent rapid nutrient losses in runoff, and hence reduce the risk of eutrophication, and thus, to minimize the environmental impacts of the poultry industry [4–8]. Due to food safety and

environmental concerns regarding PL application on agricultural land in unmodified forms, in addition to concerns regarding PL production, the importance and application of PL biochar from PL can help to partially achieve some sustainable development goals (SDGs) in order to protect the world from the threat of a continually degrading environment. In particular, it helps to achieve goals 3 (Good health and wellbeing), 6 (Clean water and sanitation), 11 (Sustainable cities and communities), 12 (Responsible consumption and production), 13 (Climate action), and 15 (Life on land) of the SDGs.

In an acidic (pH 4.5), low-soil organic C, hard-setting A horizon of Alfisols, Chan et al. [4] indicated that significant, but different changes in the chemical and physical properties of the soil were observed for both PLBs (slow pyrolysis at 450 °C and 550 °C, the latter was activated using high temperature steam), including increases in C, N, K, calcium (Ca), pH, soluble salts (SS), and available P, but a reduction in soil strength was also observed. Compared to hardwood bedding biochar, K, P, Ca, sodium (Na), and magnesium (Mg) contents were relatively high in the PLBs, and thus, the PLBs could potentially be used as either a soil conditioner or a slow-release fertilizer [7]. Due to the increase in plant growth and plant nutrients, such as N, P, Mn, Zn, Cu concentrations in soil and exchangeable cations (K, Ca, and Mg) in soil, poultry manure biochar can be effectively used for agricultural purposes [8]. Moreover, PLB can be used in crop cultivation and environmental applications in order to improve soil fertility, to enhance C sequestration, to reduce greenhouse gas emissions, and to remedy pesticides and heavy metal-contaminated soils [9]. Feedstock and process conditions during pyrolysis on the properties of PLB are important, and hence, the soil amendment values of PLBs are as well [4]. Compared to fast pyrolysis, slow pyrolysis, in general, tends to produce biochars with greater N, S, available P, Ca, Mg, surface area and cation exchange capacity (CEC). In addition, an increasing pyrolysis temperature generally tends to decrease biochar yield, but increases the biochar total C, K and Mg content, pH (ash content) and surface area, and decrease CEC [10]. During pyrolysis, thermochemical conversion changed the constitution of the biochar's carbon, the degree of aromaticity increased, and polarity reduced significantly with the pyrolysis temperature [11]. The oxidation of aromatic C occurred with the increasing pyrolysis temperature, along with the formation of carboxyl groups. Thus, the CEC decreased [12].

In two Ultisols (sandy loam-textured with pH 7.10 and silt loam-textured with pH 7.14), the biochar made by fast pyrolysis of broiler chicken litter mixed with a pine shaving base at a pyrolysis temperature of 450 °C had many beneficial effects on soil properties, including improving the water-holding capacity (WHC), cation exchange capacity (CEC), bulk density (BD), and nutrient status [13]. However, as the biochar significantly increased the contents of soluble salts (SS), pH, and extractable P, thereby exceeding agronomic levels through its higher rates, it is thus suggested that these are limiting factors that should be taken into account when determining the appropriate rate of application [13,14]. For small-scale and farm-based biochar production, slow pyrolysis is suitable due to its simplicity, efficiency and low cost [15]. Poultry litter slow pyrolysis at 300 °C should be selected for agricultural-use biochar on farms due to the higher biochar yield, total N content, OC content, and CEC value; PL slow pyrolysis at 500 °C is recommended for C sequestration and other environmental applications due to the higher C stability, BET surface area, pH, and EC [15]. Additionally, Guo et al. [9] suggested that slow pyrolysis at 300–500 °C is recommended in order to produce PL biochar for soil applications at 20–60 tons ha$^{-1}$. Additionally, PLB may serve as a sustainable, long-term source of P for soil amendments with a reduced impact on water quality and aquatic ecosystems. Furthermore, adjustments are needed to incorporate the effects of soil pH in order to determine the agronomic application practices (e.g., rate and frequency) for PLB [5]. In slightly acidic soil (pH 6.45), the OC, N, P, K, Ca and Mg of PLB-treated (300 °C) soil were higher than those of PL-treated soil [6]. However, compared to corn biochars, the higher phytotoxicity of PLB, slow pyrolyzed at 400, 500, and 600 °C, may exert a negative effect, at least at a relatively high level of soil amendment (40 t ha$^{-1}$), due to the presence of

water-soluble and biodegradable components that are probably derived from the thermal decomposition of proteins and lipids that are removable by water leaching or microbial treatments [16].

The land application of PL in open-field and greenhouse cultivation, due to bad odors and potential environmental and public health risks, has been gradually restricted since 2010 in Taiwan [17]. Thus, the production of stable PLBs from PL and its use in soil can play a vital role in solving the current problem in agriculture and could help mitigate rising greenhouse gas emissions, in addition to helping to partially achieve some SDGs. If the pyrolysis of PL is to become more widespread in Taiwan or elsewhere, it is important to understand how biochar, made from PL, affects soil properties. However, to date, the effect of PLB on soil properties are uncertain and limited and has been less studied in regards to the carbon mineralization kinetics of PLB-amended acid soil using different temperature pyrolysis and addition rates, especially it is for the applied C (PLB) mineralization. The aims of this study were: (1) to identify the optimal pyrolysis temperature for converting PL into biochar by comparing the quality characteristics of slow pyrolysis PLB at different temperatures, and (2) to compare the effects of pyrolysis temperature and rate on C mineralization kinetics in a rural acid soil when amended with PLB. The data obtained in the present study were then used to estimate the potential of such combinations of the PLB type and rate in order to preserve/restore the soil C content relative to the control. The relationships between the characteristics of each PLB and the C-mineralization dynamics were also examined.

## 2. Materials and Methods

### 2.1. Studied Soil and Biochar

Bulk soils samples (0–15 cm depth), obtained from alluvial soil developed from non-calcareous alluvial parent material from sandstone and shale, were collected from a paddy field located in Changhua Prefecture in central Taiwan. The raw PL, obtained from broiler litter manure with rice husk bedding, was collected from local broiler farms. Before pyrolysis, the loose PLs were granulated and pelleted into short cylindrical shapes (diameter < 0.3 mm, height < 1 cm). The pelleted PLs were carbonized by a horizontal and continuous carbonization furnace without the presence of oxygen, and were slow pyrolyzed at different temperatures, namely 200, 300, 400, 500, and 600 °C, to achieve a residential time of 30 min at a heating rate of 10 °C min$^{-1}$, namely, P2, P3, P4, P5, and P6 (Biomass Energy Research Lab., Department of Forestry, National Chung Hsing University, Taichung, Taiwan). The products (solid/liquid/gas) were (87%/12%/1%) for P2, (83%/15%/2%) for P3, (64%/28%/8%) for P4, (40%/41%/19%) for P5, and (37%/36%/27%) for P6. The biochar yields evidently declined after PL pyrolyzed at 400 °C, and similar yields were obtained at 500 °C and 600 °C. These biochars were used as supplied, without prior washing to remove soluble salts. For further analysis, the PLBs were homogenized and ground (<2 mm).

### 2.2. Analysis of Soil and Biochars

The studied soil characteristics, including soil pH, electrical conductivities (EC), the soils' particle sizes, total soil C contents (TC), exchangeable bases (K, Na, Ca, and Mg), cation exchangeable capacities (CEC), and base saturation percentages (BS%) were determined. The analysis methods used were the same as those described in previous studies [17,18]. The characterizations of the five studied PLBs, including the pH; EC; available N concentration; CEC; plant-available nutrients (P, K, Ca, and Mg); total contents of P, K, Ca, and Mg; elemental analyses (C, N, H, O, and S); and $^{13}$C-Nuclear Magnetic Resonance (NMR) spectra, were determined, and the analysis method used was the same as that described in a previous study [17,18]. Solid-state, cross-polarization magic angle-spinning and total-sideband-suppression $^{13}$C-NMR spectra were obtained using a Bruker Avance III 400 NMR spectrometer (Bruker Corporation, Billerica, MA, USA) at 100 MHz (400 MHz $^1$H frequency). The additional parameters used to acquire and process the

spectra were as follows: a contact time of 1.5 ms, a recycle delay time of 1 s, a line broadening of 15 Hz, a spinning speed of 10 kHz, and a scan number of 8000. The chemical shift between 0 and 50 ppm was assigned to paraffinic C; 50–109 ppm was attributed to substituted aliphatic C, including alcohol, amines, carbohydrates, ethers, and methyl and acetal C; 109–145 ppm was attributed to aromatic C; 145–163 ppm was attributed to phenolic C; 163–190 ppm was attributed to carboxyl; and 190–220 ppm was attributed to carbonyl carbons [17]. Additionally, extraction of water-soluble biochar C and N was also conducted. This extraction process, which involved gentle shaking with deionized water (water/biochar ratio 10:1) for 30 min, was repeated 5 times for each biochar sample with three replicates [19–21]. Water extracts were filtered through 0.45 mm pore-size nylon membrane filters (Whatman®, Maidstone, UK) and were collected. The pH and EC of the extraction, in addition to water-extractable organic C (WEOC), water-extractable $NO_3^-$–N and $NH_4^+$–N concentrations, were determined using the analysis methods described in previous studies [20,21].

### 2.3. Soil-Biochar Incubation Experiment

For each PLB, three treatments with three replicates were conducted in the incubation experiment: (1) soil + 1% biochar (P2-1, P3-1, P4-1, P5-1, and P6-1); (2) soil + 5% biochar (P2-5, P3-5, P4-5, P5-5, and P6-5); and (3) soil + 10% biochar (P2-10, P3-10, P4-10, P5-10, and P6-10). The control was the natural soil + 0% biochar (three replicates). Application rates of 5% and 10% were chosen to help determine which higher levels of biochar application are harmful to the soil. We must confess that the application of these high rates (5% and 10%, by wt.) of biochar may be problematic in practice, e.g., difficult for operational incorporation in the field, but this is not the focus of the present work. Soil and biochar were added to each jar and a spoon was used to thoroughly mix the samples. In a similar way to a previous study [17,18,20,21], 25 g of the mixed soil sample was incubated in a 30 mL plastic container inside a 500 mL plastic jar, with 10 mL of 1 M NaOH and 10 mL of water in a vial added separately. Three blanks consisting of jars with only water and NaOH, as described above, were also included. The jars were sealed with a rubber bung and incubated in a randomized block design at 25–27 °C for 56 days. The soil moisture content was adjusted to 60% of the field capacity before incubation and was maintained throughout the experiment using repeated weightings. The NaOH vials were changed after 0, 1, 2, 3, 5, 9, 17, 30, and 56 days in order to determine the evolved $CO_2$ [22]. Soil respiration data was reported as the mg of the $CO_2$–C respired per kilogram of soil.

### 2.4. Statistical Analysis

The amount of $CO_2$–C released during different time intervals in each treatment was used to conduct the cumulative $CO_2$–C released and the C mineralization kinetics. The first-order exponential model and the double exponential model were mostly used to describe the C mineralization in treated and non-treated soils [23]. However, according to the report [24], and a previous study [20], the double exponential model was used in the present work to mathematically describe the biochar-amended soil C degradation, assuming that there were two C pools: a rapidly degrading C-pool and a slowly degrading or recalcitrant C-pool:

$$C_t = C_l \times (e^{-k_l t}) + C_r \times (e^{-k_r t}) \tag{1}$$

The half-life of C in biochar-amended soil is:

$$t_{1/2} = \ln(2)/k \tag{2}$$

The mean residence time (MRT) of C in biochar-amended soil is [25]:

$$MRT = 1/k \tag{3}$$

where $C_l$ and $C_r$ indicate the amount of potentially mineralizable C in the labile and resistant fractions (%), respectively; $k_l$ and $k_r$ are the respective mineralization rate constants ($d^{-1}$); and t is time (d).

The cumulative mineralized C (CMC) at time t was calculated in the incubation, based on the amount of $CO_2$–C that evolved in each treatment, and the amount of applied C mineralized (ACM) was also determined as [23]:

$$\% \text{ ACM} = [(\text{CMC}_{treatment} - \text{CMC}_{control})/\text{organic C applied}] \times 100 \tag{4}$$

This approach assumes that there was no priming effect [26]. Additionally, two kinetic models were used to describe the ACM data in the different treatments:

First-order kinetic model [27]:

$$C_t = C_0 \times (1 - e^{-kt}) \tag{5}$$

where $C_0$ is the potentially mineralizable C (%) and k the mineralization rate constant ($d^{-1}$);

Double exponential model [28]:

$$C_t = C_l \times (1 - e^{-k_l t}) + C_r \times (1 - e^{-k_r t}) \tag{6}$$

where $C_l$ and $C_r$ indicate the amount of potentially mineralizable C in the labile and resistant fractions (%), respectively; $k_l$ and $k_r$ are the respective mineralization rate constants ($d^{-1}$); and t is time (d).

A nonlinear regression using a double exponential model (Sigma-plot 14.5, tolerance $1 \times e^{-10}$, step size 100, and 1200 iterations; Systat Software, Inc., San Jose, CA, USA) was performed to mathematically define the size and turnover rate of $C_1$, which corresponds to a small and easily mineralizable C pool with a high turnover rate ($k_l$), and $C_r$ corresponds to a large, stable pool with a low turnover rate ($k_r$) consisting of stable C. The low turnover rate value ($k_r$) was used to calculate the half-life ($t_{1/2}$) of the most stable C fraction using Equation (2). In the ACM curve, the mineralization rate constant ($d^{-1}$), the half-life of C in soil, and MRT were also calculated. The single and double exponential equations were compared based on the adjusted $r^2$. The difference between $CO_2$–C evolved between biochar-amended treatments and the un-amended control was calculated to examine the percentage of decline or increase due to biochar addition [29].

The Statistical Analysis System (SAS) 9.4 package (SAS Institute Inc., Cary, NC, USA) was used for statistical analyses, including the calculation of means and standard deviations, and the differences between the means. Arithmetic means of the evolved $CO_2$–C and ACM were calculated from each consecutive measurement date. A repeated measure multivariate analysis of variance (MANOVA) was used to test the biochar pyrolyzed temperature, addition rates, and their interaction on evolved $CO_2$–C for each incubation period. The results were analyzed by one-way ANOVA in order to test the effects of each treatment. Significantly different means were compared via the least significant difference (LSD) based on a *t*-test at a 5% probability level.

## 3. Results

### 3.1. Characterization of Initial Soil and Poultry Litter Biochar (PLB)

The characterizations of the studied soil include acidic soil pH (pH 5.96), clay loam soil texture (26.0% sand, 40.9% silt, and 33.1% clay), EC 2.13 dS m$^{-1}$, total C 2.16%, exchangeable bases (K/Na/Ca/Mg) 0.27, 2.38, 2.94, and 0.82 cmol (+) kg$^{-1}$ soil$^{-1}$, CEC 15.7 cmol (+) kg$^{-1}$ soil$^{-1}$, and a base saturation 41.0%. The pH of the studied PLBs ranged from 6.94 (P2) to 9.81 (P6) and increased with increasing pyrolyzed temperatures (Table 1).

**Table 1.** Characteristics of the five studied biochars.

| Characteristics [1] | P2 [2] | P3 | P4 | P5 | P6 |
|---|---|---|---|---|---|
| pH [3] | 6.94 | 6.83 | 7.12 | 9.66 | 9.81 |
| EC (dS m$^{-1}$) | 9.18 [3]/9.23 [4] | 9.33 [3]/9.52 [4] | 8.50 [3]/10.5 [4] | 8.69 [3]/12.4 [4] | 8.71 [3]/12.5 [4] |
| Available N (g kg$^{-1}$) | 11.4 | 10.6 | 5.65 | 0.94 | 0.39 |
| CEC (cmol(+) kg$^{-1}$) | 37.5 | 33.2 | 21.9 | 31.5 | 29.3 |
| M3-P (g kg$^{-1}$) | 7.95 | 6.32 | 6.46 | 8.86 | 6.38 |
| M3-K (g kg$^{-1}$) | 27.9 | 24.3 | 25.4 | 42.9 | 35.1 |
| M3-Ca (g kg$^{-1}$) | 8.40 | 6.96 | 7.48 | 10.1 | 8.29 |
| M3-Mg (g kg$^{-1}$) | 5.45 | 4.52 | 4.42 | 5.76 | 4.19 |
| Total P (g kg$^{-1}$) | 9.28 | 11.6 | 13.4 | 20.9 | 19.8 |
| Total K (g kg$^{-1}$) | 24.0 | 28.8 | 33.4 | 49.4 | 47.9 |
| Total Ca (g kg$^{-1}$) | 17.8 | 22.8 | 26.2 | 40.3 | 38.3 |
| Total Mg (g kg$^{-1}$) | 6.14 | 7.45 | 8.73 | 13.8 | 13.0 |
| C% | 36.4 | 39.1 | 41.0 | 46.4 | 46.3 |
| N% | 3.17 | 3.19 | 3.53 | 3.55 | 3.39 |
| H% | 5.42 | 5.11 | 4.89 | 2.74 | 2.65 |
| O% | 41.5 | 39.1 | 34.6 | 17.4 | 17.1 |
| S% | 1.10 | 1.08 | 1.18 | 1.84 | 1.82 |
| (O+N)/C atomic ratio | 0.93 | 0.82 | 0.71 | 0.35 | 0.34 |
| O/C atomic ratio | 0.76 | 0.67 | 0.56 | 0.25 | 0.25 |
| H/C atomic ratio | 1.79 | 1.57 | 1.43 | 0.71 | 0.69 |

| C Type (%) Distribution of C Chemical Shift (ppm) (Integrated Results of Solid-State $^{13}$C NMR Spectra) | Raw | P2 | P4 | P6 |
|---|---|---|---|---|
| Paraffinic C (0–50 ppm) | 12 | 14 | 20 | 25 |
| Substituted aliphatic C including alcohol, amines, carbohydrates, ethers, and methyl and acetal C (50–109 ppm) | 76 | 71 | 57 | 17 |
| Aromatic C (109–145 ppm) | 4.4 | 5.7 | 11 | 35 |
| Phenolic C (145–163 ppm) | 3.7 | 4.3 | 5.7 | 10 |
| Carboxyl C (163–190 ppm) | 3.5 | 5.1 | 4.8 | 4.9 |
| Carbonyl C (190–220 ppm) | 0.2 | 0.3 | 1.4 | 7.8 |
| Aliphatic C [5] | 88 | 85 | 77 | 42 |
| Polar C [5] | 84 | 80 | 69 | 40 |
| Aliphatic polar C [5] | 76 | 71 | 57 | 17 |

[1] P2, P3, P4, P5, P6 = poultry litter pyrolyzed at 200, 300, 400, 500, and 600 °C; [2] EC = electrical conductivity; CEC = cation exchange capacity; M3 = Mehlich–3 extractable; [3] The pH and EC of the biochar were measured using 1:5 solid; the solution ratio after shaking for 30 min in deionized water was calculated; [4] Biochar EC was measured after shaking the biochar–water mixtures (1:5 solid: solution ratio) for 24 h; [5] Aliphatic C: total aliphatic C region (0–109 ppm); polar C: total polar carbon region (50–109 ppm and 145–220 ppm); aliphatic polar C: polar carbon in the aliphatic region (50–109 ppm).

The EC values were very high (>8.5 dS m$^{-1}$) and decreased with pyrolyzed temperature increases after shaking for 30 min and increased with pyrolyzed temperature increases after shaking for 24 h. The available N sharply reduced after being pyrolyzed at by 500 and 600 °C, indicating the very low number of available N applications. P2 had the highest CEC value, followed by P3, P5, P6, and P4, and the values ranged from 21.9 to 37.5 cmo (+) kg$^{-1}$. In addition, the content of available nutrients (P, K, Ca, and Mg) by Mehlich–3 extraction was the highest in P5 and was the lowest in P3. The content of available K was higher than P, Ca, and Mg, which could be due to an abundance of rice husk or rice straw in the raw material of PL—both are popular materials for chicken farmers in Taiwan. The total contents of P, K, Ca, and Mg showed a similar trend—that is, increasing with an increasing pyrolyzed temperature and the highest content being in P5. Due to the dehydration and decarboxylation reaction, the atomic ratios of hydrogen–oxygen (H/C) and oxygen–carbon

(O/C) showed a sharp decrease with the increase in the pyrolysis temperature (Table 1). The C type and the integrated results of the solid-state $^{13}$C–NMR spectra (%) further support the results described above (Table 1), and aromatic C significantly increased with increasing temperature. In contrast, the aliphatic C, polar C, and aliphatic polar C both significantly decreased.

### 3.2. Water Soluble Extracts of Five Poultry Litter Biochars

After five washings, the solution pH of the P2, P3, and P4 biochars were similar: pH 7.63, 7.53, and 7.50, respectively, but was higher in P5 and P6 (pH 8.75 and pH 9.23; Figure 1a). The difference in the solution pH between the first and fifth washing was the highest (0.76 pH units) in P2. The solution EC values sharply declined during the first and second washing, and then gradually declined towards the final wash (Figure 1b).

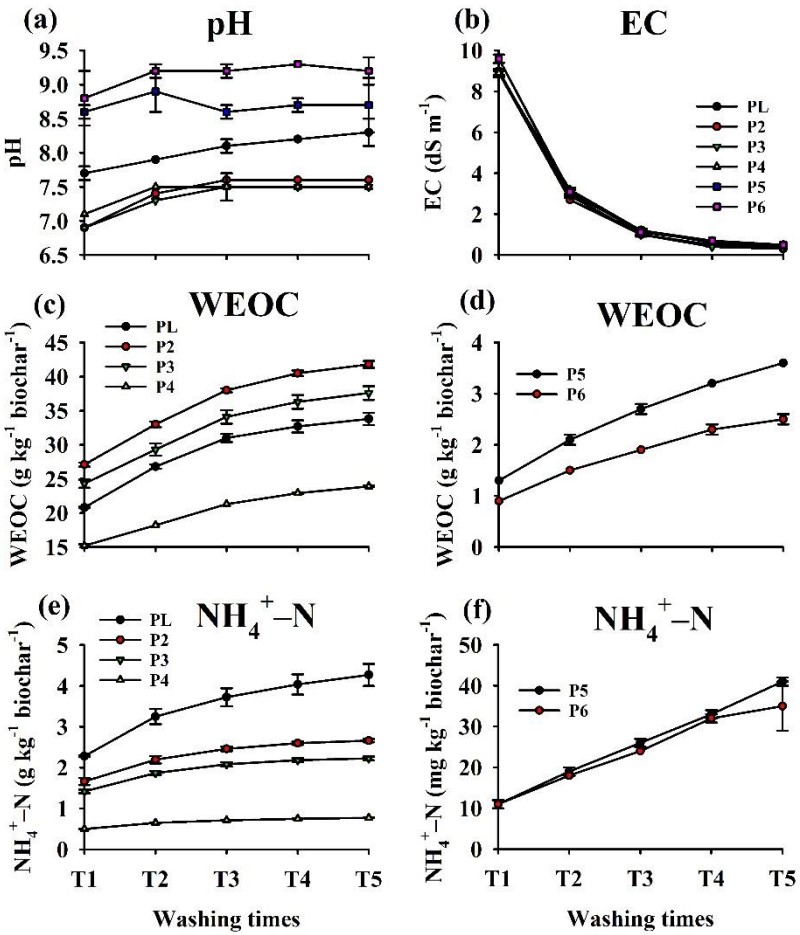

**Figure 1.** The values of (**a**) pH and (**b**) electrical conductivity (EC), and concentrations of (**c**,**d**) cumulative organic C and (**e**,**f**) ammonium nitrogen (NH$_4^+$–N) extracted from PL, P2, P3, P4, P5, and P6 (PL pyrolyzed at 200 °C, 300 °C, 400 °C, 500 °C, and 600 °C, respectively) biochar by water 5 times. Error bars represent standard deviations from the mean (*n* = 3).

After five washings, the EC value of raw PL and five biochars was similar, ranging from 0.27 to 0.53 dS m$^{-1}$. The difference in the solution EC between the first and fifth washing was the highest (9.10 dS m$^{-1}$) in P6 biochar. Most of the WEOCs were extracted after five washings, and the cumulative WEOCs evidently decreased with an increasing pyrolysis temperature (Figure 1c,d). The cumulative WEOC value was 33.8, 41.8, 37.6, 23.9, 3.63, and 2.55 g C kg$^{-1}$ biochar for PL, P2, P3, P4, P5, and P6, respectively. In a similar way to the change in WEOCs, five washings extracted most of the water that was extractable N, most of the ammonium (NH$_4^+$–N), and most of the cumulative N evidently decreased with

the increasing pyrolysis temperature (Figure 1e,f). Except for the PL, the water extraction nitrate was almost under-detected for all PLBs (data not shown). However, five washes extracted most of the water-extractable $NH_4^+$–N, and the results of the cumulative $NH_4^+$–N was 4.23, 2.66, 2.23, and 0.77 g $kg^{-1}$ $biochar^{-1}$ for PL, P2, P3, and P4 biochar, and 41 and 35 mg $kg^{-1}$ $biochar^{-1}$ for P5 and P6 biochar, respectively.

### 3.3. Soil Carbon Mineralization

As shown in Figure 2, the current study indicated that PLB treatments had significant effects on the soil respiration. Additionally, the $CO_2$–C release significantly varied with the treatments, rate, incubation time, and all the interactions between these factors ($p < 0.0001$). The temporal $CO_2$–C release increased rapidly during the 0 to 2 days incubation periods and was the highest on day 2 for the 5% and 10% addition of P2 and P3, but was day 1 for P4-5 and P4-10 treatments (Figure 2b). Soil respiration quickly declined during the 2 to 30 days incubation period for the 10% addition of P2 and P3, but was only one day (the 2 to 3 day) for the 5% addition of P2 and P3. In addition, soil respiration gradually changed thereafter to the end of the incubation for all the treatments, and it possibly reached a plateau. For the 1% addition of P2, P3, and P4, and 1%, 5%, and 10% of P5 and P6, soil respiration increased rapidly between the 0 to 1 day incubation period, it was the highest at day 0, it obviously declined between 0 to 3 days, and gradually increased thereafter to the end of the incubation. Except for P5-10, the temporal changes during the 9 to 56 days incubation period may have reached a plateau. After the initial $CO_2$–C emission flushing, the mineralization rate of C in all the treatments was significantly reduced, especially P2 and P3 biochar. Therefore, in the current work, C mineralization may have occurred in two main stages: the first initial burst, in which easily decomposed C was released quickly, and the second burst, in which the more stubborn C was degraded.

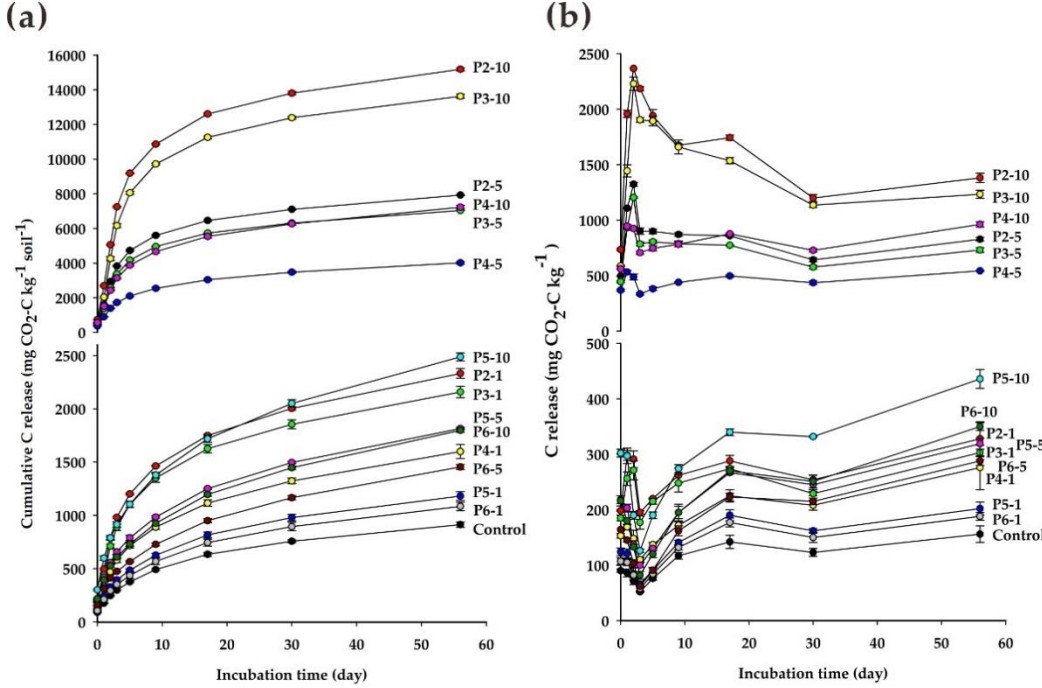

**Figure 2.** (**a**) Cumulative and (**b**) temporal $CO_2$–C release for all treatments from studied soil during the incubation period. P2, P3, P4, P5, and P6 is the poultry litter pyrolyzed at 200, 300, 400, 500, and 600 °C, respectively; −1 = 1% biochar addition; −5 = 5% biochar addition; and −10 = 10% biochar addition. Values represent means (*n* = 5) ± standard deviation (error bars).

In the present work, the amount of $CO_2$-C released at all the monitoring times in the incubation significantly increased as the addition rate increased (Tables S1 and S2). The

release of $CO_2$–C at all the monitoring times of incubation was significantly the highest in P2, followed by P3, P4, P5, P6, and the control. At the end of the incubation period, the largest cumulative soil $CO_2$–C release occurred in the P2-10 treatment (15,187 mg $CO_2$–C $kg^{-1}$ $soil^{-1}$), and significantly increased, by 1562%, as compared to the control (Table S3). The P5-10 and P6-10 treatments also showed significant increases, approximately 172% and 97%, respectively. The labile C pool of the P2 and P3 treatments evidently increased with the increasing addition rate, as compared to the control, and there was a less evident increase in the P4 treatment and few changes in the P5 and P6 treatments (Table 2). The first stage of the P5 and P6 treatments, including the 1%, 5%, and 10% additions, was shorter (1.85~2.55%) than the control (3.75%), due to the relatively less-mineralizable C pool ($C_l$) and the much higher rate constants ($k_l$). In addition, except for the control and P6-5 treatment, the half-life ($t_{1/2}$) and mean residence time (MRT) in the labile C pool were less than 3 days and 4 days, respectively, and there were only few changes with the increased addition rate. The short half-life and MRT in the labile C pool were consistent with the temporal changes in soil respiration (Figure 2b), which indicates an abrupt change during the 0 to 3 days of incubation periods. During the second stage of C mineralization, the pool of relatively stable C ($C_r$) in P2, P3, and P4 treatments decreased with the increasing addition rate, and on the contrary, the rate constants ($k_r$) showed a slight increase. In the treatments of P5 and P6, the pool of $C_r$ was similar in regards to the addition rate, but the rate constants ($k_r$) showed a slight decrease with the increasing addition rate.

**Table 2.** Kinetic parameters [1] of carbon mineralization.

| Treatment [2] | Labile C Pool | | | | Resistant C Pool | | | | Rsqr | Adj Rsqr |
|---|---|---|---|---|---|---|---|---|---|---|
| | $C_l$ (%) | $k_l$ (% $d^{-1}$) | $t_{1/2}$ (d) | MRT (d) | $C_r$ (%) | $k_r$ (% $d^{-1}$) | $t_{1/2}$ (Year) | MRT (Year) | | |
| Control | 3.75 | 0.08 | 9 | 12 | 95.9 | – [3] | – | – | 0.962 | 0.939 |
| P2-1 | 5.41 | 0.24 | 3 | 4 | 94.6 | 0.0005 | 4 | 5 | 0.998 | 0.997 |
| P2-5 | 15.0 | 0.32 | 2 | 3 | 84.9 | 0.0014 | 1 | 2 | 0.996 | 0.994 |
| P2-10 | 22.0 | 0.28 | 2 | 4 | 78.1 | 0.0015 | 1 | 2 | 0.995 | 0.992 |
| P3-1 | 4.22 | 0.56 | 1 | 2 | 95.7 | 0.0009 | 2 | 3 | 0.994 | 0.990 |
| P3-5 | 12.9 | 0.37 | 2 | 3 | 87.4 | 0.0012 | 2 | 2 | 0.996 | 0.993 |
| P3-10 | 18.3 | 0.27 | 3 | 4 | 81.8 | 0.0014 | 1 | 2 | 0.998 | 0.997 |
| P4-1 | 2.90 | 0.41 | 2 | 2 | 97.1 | 0.0006 | 3 | 5 | 0.985 | 0.976 |
| P4-5 | 6.65 | 0.22 | 3 | 4 | 93.0 | 0.0006 | 3 | 5 | 0.991 | 0.986 |
| P4-10 | 7.92 | 0.29 | 2 | 3 | 91.8 | 0.001 | 2 | 3 | 0.994 | 0.990 |
| P5-1 | 1.82 | 0.55 | 1 | 2 | 98.2 | 0.0006 | 3 | 5 | 0.971 | 0.953 |
| P5-5 | 1.99 | 0.50 | 1 | 2 | 98.0 | 0.0004 | 5 | 7 | 0.958 | 0.933 |
| P5-10 | 2.05 | 0.23 | 3 | 4 | 97.8 | 0.0003 | 6 | 9 | 0.962 | 0.939 |
| P6-1 | 2.05 | 0.23 | 3 | 4 | 97.8 | 0.0003 | 6 | 9 | 0.962 | 0.939 |
| P6-5 | 2.55 | 0.08 | 9 | 12 | 97.0 | – | – | – | 0.919 | 0.871 |
| P6-10 | – [3] | – | – | – | – | – | – | – | – | – |

[1] $C_r$ = 100-$C_l$; Parameters of the double-exponential equation were obtained by nonlinear regression using Sigma plot 14.0.; $k_l$, rate constant for relatively easily mineralizable C pool ($C_l$); $k_r$, rate constant for slowly mineralizable C pool ($C_r$); [2]: P2, P3, P4, P5, P6 = poultry litter pyrolyzed at 200, 300, 400, 500, and 600 °C; −1 = 1% biochar addition; −5 = 5% biochar addition; −10 = 10% biochar addition; [3] − = low reliability results.

The half-life and MRT of $C_r$ in biochar-amended soil, calculated based on the slow reaction rate constant ($k_r$), were less than six and nine years, respectively. Higher temperature pyrolysis PLBs, such as P4, P5, and P6, could have a higher potential to sequestrate C, as can be seen in the current study.

### 3.4. Amount of Applied Carbon Mineralized

The curve patterns of ACM (Figure 3) were consistent with the WEOCs (Figure 1c,d) and water extractable N ($NH_4^+$−N) (Figure 1e,f). That is, the pyrolysis temperature could clearly be separated into three groups: low temperature (≤300 °C, P2 and P3), intermediate temperature (400 °C, P4), and high temperature (>500 °C, P5 and P6). ACM regularly increased up to day 30 (P5-1, P5-5, P5-10, P6-1, P6-5, and P6-10 treatment), where it reached a plateau, indicating that soil microbial activity stabilized. However, ACM at the end of the incubation still showed an obvious increase (P2-1, P2-5, P2-10, P3-1, P3-5, and P3-10

treatments) or a gradual increase (P4-1, P4-5, and P4-10 treatments), indicating that the soil microbial activity was not stabilized. These results indicate that the microbial degradation time for organic matter in these PLB-amended soils was prolonged, relative to the basic level of microbial activity in unamended soils. At the end of the incubation (56 days), the P2-10 treatment had the highest ACM values (42.2% or 422 mg (g biochar C applied)$^{-1}$), and the lowest was in the P6-10 treatment (2.06% or 20.6 mg (g biochar C applied)$^{-1}$).

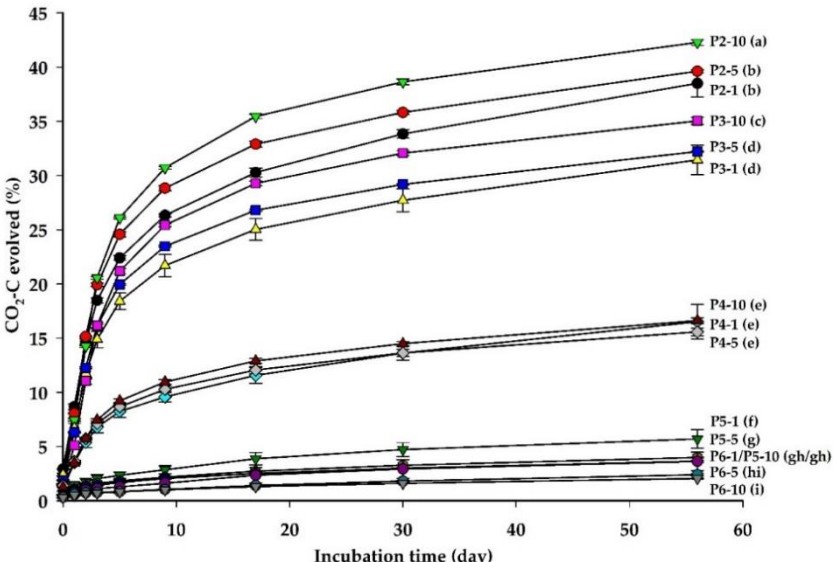

**Figure 3.** Cumulative amounts of applied C mineralized (ACM) evolved as $CO_2$–C in the different treatments during the incubation. Values represent means ($n = 5$) $\pm$ standard deviation (error bars). The different lowercase letters in parentheses indicate significant differences ($p < 0.05$) between treatments at the end of the incubation (56-d).

We used single first-order and double-exponential equations to describe ACM data. The $r^2$ values indicate the degrees of conformity between the experimental data and the equation-predicted values. The double-exponential equation describes the ACM data more accurately than the single first-order equation for all the treatments (data not shown). The kinetic parameters of ACM data calculated using the double-exponential equation suggested a biphasic C mineralization process, as previously described in regards to the C mineralization of biochar-amended soils. According to the parameters (Table 3), increasing amounts of P2 and P3 biochar led to a clear increase in the labile pool ($C_l$), while the recalcitrant one ($C_r$) tended to decline, and the total potentially mineralizable C ($C_l + C_r$) was the highest with the 10% addition, but was similar between the 1% and 5% addition. In P4 treatments, the $C_l$ showed a slight increase, while the $C_r$ showed a slight decline, and the P4-1 treatment had the highest potentially mineralizable C, followed by P4-10 and P4-5. However, increasing amounts of P5 and P6 biochar led to a slight decrease in the labile pool, the recalcitrant pool, and the total potentially mineralizable C. The $C_l$ pool and $C_r$ pools in P5 (1.2~1.5% and 3~5%) and P6 (0.59~0.75% and 2~3%) treatments were much smaller than in P2 (22~30% and 16~20%) and P3 (18~26% and 13~16%) treatments. In P2 and P3 treatments, the amount of fast mineralizable C were higher than those of the recalcitrant fraction, but in P4, P5, and P6 treatments, it was the inverse. Indeed, the fast mineralizable C of the total amount of potentially mineralizable C amounted to between 52% and 64% in P2 treatments, 54% and 64% in P3, 37% and 49% in P4, 24% and 31% in P5, and 19% and 24% in P6, respectively.

**Table 3.** Model parameters and coefficients of determination ($r^2$) estimated using the double exponential model fitted to the applied C mineralized (ACM) data from the different treatments.

| Treatment [1] | Labile C Pool | | | | Recalcitrant C Pool | | | | C Pool $(C_l+C_r)$(%) | $C_l$ (% $(C_l+C_r)$) | Rsqr | Adj Rsqr |
|---|---|---|---|---|---|---|---|---|---|---|---|---|
| | $C_l$ (%) | $k_l$ (% d$^{-1}$) | $t_{1/2}$ (d) | MRT (d) | $C_r$ (%) | $k_r$ (% d$^{-1}$) | $t_{1/2}$ (d) | MRT (d) | | | | |
| P2-1 | 21.7 | 0.475 | 1 | 2 | 20.3 | 0.031 | 22 | 32 | 42.0 | 51.7 | 0.992 | 0.987 |
| P2-5 | 26.1 | 0.386 | 2 | 3 | 16.7 | 0.029 | 24 | 34 | 42.9 | 61.0 | 0.995 | 0.992 |
| P2-10 | 29.5 | 0.321 | 2 | 3 | 16.6 | 0.026 | 27 | 38 | 46.1 | 64.0 | 0.996 | 0.993 |
| P3-1 | 18.4 | 0.431 | 2 | 2 | 16.0 | 0.030 | 23 | 34 | 34.4 | 53.6 | 0.992 | 0.987 |
| P3-5 | 21.4 | 0.377 | 2 | 3 | 13.4 | 0.029 | 24 | 34 | 34.8 | 61.5 | 0.995 | 0.992 |
| P3-10 | 25.9 | 0.275 | 3 | 4 | 14.7 | 0.017 | 40 | 57 | 40.7 | 63.7 | 0.995 | 0.993 |
| P4-1 | 7.19 | 0.569 | 1 | 2 | 12.4 | 0.025 | 28 | 40 | 19.6 | 36.7 | 0.987 | 0.980 |
| P4-5 | 7.74 | 0.537 | 1 | 2 | 8.85 | 0.038 | 18 | 26 | 16.6 | 46.7 | 0.989 | 0.982 |
| P4-10 | 8.85 | 0.444 | 2 | 2 | 9.25 | 0.032 | 21 | 31 | 18.1 | 48.9 | 0.993 | 0.988 |
| P5-1 | 1.46 | 1.969 | 0.4 | 1 | 4.70 | 0.041 | 17 | 25 | 6.16 | 23.7 | 0.976 | 0.961 |
| P5-5 | 1.22 | 1.182 | 1 | 1 | 2.76 | 0.037 | 20 | 29 | 3.98 | 30.6 | 0.965 | 0.944 |
| P5-10 | 1.33 | 1.273 | 1 | 1 | 3.09 | 0.035 | 19 | 27 | 4.42 | 30.0 | 0.969 | 0.950 |
| P6-1 | 0.75 | 2.054 | 0.3 | 0.5 | 3.29 | 0.037 | 19 | 27 | 4.04 | 18.6 | 0.984 | 0.975 |
| P6-5 | 0.62 | 2.195 | 0.3 | 0.5 | 2.60 | 0.021 | 34 | 49 | 3.22 | 19.3 | 0.968 | 0.948 |
| P6-10 | 0.59 | 1.480 | 0.5 | 1 | 1.85 | 0.028 | 25 | 36 | 2.45 | 24.2 | 0.967 | 0.946 |

[1]: P2, P3, P4, P5, and P6: poultry litter pyrolyzed at 200, 300, 400, 500, and 600 °C, respectively; −1 = 1% biochar addition; −5 = 5% biochar addition; −10 = 10% biochar addition.

The half-life ($t_{1/2}$) and mean residence time (MRT) in the labile C pool only showed a few changes between the three rates and was less than 3 days and 4 days in P2, P3, and P4 treatments, and less than 1 day in P5 and P6 treatments (Table 3). These results were similar to the findings in the C mineralization of PLB-amended soils (Figure 2), suggesting that this the most labile C fraction is broken down within a short time. Based on the slow reaction rate constant ($k_r$), the changes in the $t_{1/2}$ and MRT were similar to the potentially mineralizable C and showed a decreasing order of 10% > 1%~5% in both P2 and P3 treatments, 1% > 10% > 5% in P4 treatment, and 5% > 10% > 1% in both P5 and P6 treatments. The P3-10 treatment had the longest half-life and MRT, 40 days and 57 days, respectively, and the lowest was for P5-1 treatment, 17 days and 25 days, respectively. However, the $C_r$ pool of P5-1 treatment was about one of third compared to the P3-10. The lower O/C atomic ratio and higher aromatic C (Table 1) in these high temperature pyrolysis PLBs suggest that these Cs are highly recalcitrant.

## 4. Discussion

### 4.1. Quality Characteristics of Poultry Litter Biochar

Elemental analysis (Table 1) indicated that the total C content of PLBs increased with increasing temperature, indicating a carbonization degree increment with an increasing temperature [11]. The total C, N, and sulfur contents of PLBs were between 33.7% and 46.4%, 2.94% and 3.55%, and 1.08% and 1.84%, respectively, which is higher than that reported by Pariyar et al. [11]. Those differences could be attributed to the different compositions of raw PLs. The current PLBs could have the potential to supply additional N fertility.

The PLB had a pH in the range of 6.8–9.8 (Table 1), and this increased with the pyrolysis temperature, but the values were relatively lower than that reported by Song and Guo [15] (pH 9.5–11.5) and Pariyar et al. [11] (pH 6.3–10.1). This was probably due to the feedstock composition and use of the measurement method. Consistent with the pH, the contents of K, Ca, and Mg in PLB increased with the increasing pyrolysis temperature, and this confirmed the suggestions by Dodson et al. [30]. Using high pH biochar and the presence of calcite can achieve a good neutralization effect [11]. The results of this study indicate that higher temperature pyrolysis PLBs (>500 °C) have a greater liming potential than the lower temperature PLBs (<400 °C) and a higher neutralizing effect when applied to the soil. In addition to the higher pH, the higher K, Ca, and Mg content (Table 1), including their availability and total content, of P5 and P6 support this possibility. The PLB had a

high EC value in the range of 8.5 to 9.3 dS m$^{-1}$ (30 min) and 9.2 to 12.5 dS m$^{-1}$ (24 h), and this increased gradually with the pyrolysis temperature after 24 h. The EC value of the studied PLBs were similar to that obtained by Pariyar et al. [11] (9.3–12.8 dS m$^{-1}$) but were obviously lower than that reported by Song and Guo [15] (22.8–31.0 dS m$^{-1}$). In order to avoid the potential toxicity of crop seeds and seedlings, and considering the high salinity of PLB, PLB should be applied to the soil in a controlled ratio [15]. In addition to PLB, the soil EC value was higher (2.13 dS m$^{-1}$) and this was probably due to the higher exchangeable Na and Ca contents.

The supply of available nutrients by biochar can be quite variable and even biochars obviously contain a large or excessive amount of inorganic elements [10], including both feedstock material and a pyrolysis temperature, which had an influence on the available nutrients in biochar. Temperatures of more than 760 °C are needed to vaporize K, P, S, Na, Mg, and Ca [31]. The studied PLBs were slowly pyrolyzed at a temperature lower than 760 °C, and there was an obviously higher concentration of total P, K, Ca, and Mg in the biochars (Table 1), which increased with the increasing pyrolysis temperature. A PLB with a higher P content was an obvious association between higher Mg and Ca, but was less obvious in the K content, which is consistent with previous reports [10,32], and both suggest that the coordinated cations (like Fe, Al, Ca, and Mg) play a key role in controlling the availability of P in biochars. Several reports [33–35] indicated that P availability in biochar decreased with increasing production temperatures. In this study, the available P (water extraction) was about 86, 54, 48, 42, and 32% of the total P for P2, P3, P4, P5, and P6 biochar, respectively, suggesting that the low temperature pyrolysis PLB (e.g., P2 and P3) had a higher content of P minerals that could be solubilized in acidic conditions. As an abundance of soluble P may form during the biochar process, the content of available P could increase in biochar-amended soil after the addition of biochar as a soil amendment [36].

Furthermore, the available K (water extraction) in this study was in the same range as previous reports [10,20], and was about 100%, 84%, 76%, 87%, and 73% of the total K for P2, P3, P4, P5, and P6 biochar, respectively. Cantrell et al. [37] showed that the best predictor for manure-based biochar electrical conductivity (EC) values was the total concentration of (K + Na) combined, which indicates that the form of K in the biochar was water-soluble. In addition, the available Ca and Mg of the total Ca and Mg decreased with an increasing pyrolysis temperature; for P2, P3, P4, P5, and P6, it was approximately 47% and 89%, 31% and 61%, 29% and 51%, 25% and 42%, and 22% and 32%, respectively (Table 1). In this study, the abundance of available K increased with the increasing rate and pyrolysis temperature due to it having the highest available K and total K in PLB, as compared to P, Ca, and Mg (Table 1), especially considering the high temperature PLB. In addition, an increased concentration of K in biochar, together with Mg and Ca, functions as a liming agent in order to neutralize acid soils [38].

On average, the contents of available P and K soil could be significantly increased by 32.4% and 48.3%, respectively [39]. Using P and K-rich biochars could act as nutrient suppliers and the improvers in low-P or -K soils [40,41]. In this study, the total contents of P, K, Ca, and Mg evidently concentrate in high temperature pyrolysis PLBs (P5 and P6), but K tends to be highly available in both low and high temperature pyrolysis PLBs. The current results suggested that low temperature pyrolysis PLBs (P2, P3, and P4) could play an important role in supplying available nutrients. PLBs produced at low pyrolysis temperatures (e.g., 300 °C) is rich in N, P, K, S, Ca, Mg and other plant nutrients [15]. If the PLB is used as a soil amendment, it can immediately improve fertility, indicating that PLB can be used as a slow-release nutrient amendment in crop production, thereby reducing the risk of nutrient loss after land application [15]. However, not all biochars are created equal in terms of supplying available plant nutrients and supply relevant amounts of plant nutrients, but there is a potential for all biochars to act as soil conditioners [10].

According to the recommendations from the government's agricultural authorities, Council of Agriculture, Executive Yuan, R.O.C. (Taiwan) (https://eng.coa.gov.tw; https:

//m.coa.gov.tw/) (accessed on 18 May 2021), the recommended amount of fertilizer for optimal short-term leafy crop yields is 150–250 kg of N ha$^{-1}$ (on average 200 kg ha$^{-1}$), 50–75 kg of $P_2O_5$ ha$^{-1}$ (on average 62.5 kg ha$^{-1}$) and 100–150 kg of $K_2O$ ha$^{-1}$ (on average 125 kg ha$^{-1}$). Given the availability of P and K concentrations in five studied PLBs (Table 1), it would require 3.07–4.30 Mg ha$^{-1}$ in order to meet the crop P demands and 3.85–6.80 Mg ha$^{-1}$ to meet the crop K demands. The addition rate is about 0.15–0.21% for P and 0.19–0.34% for K, respectively (assuming it has ~2000 Mg soil ha$^{-1}$). In this study, given the availability of N concentration in the studied PLBs (Table 1), approximately 18 (P2), 19 (P3), 35 (P4), 213 (P5), and 513 (P6) Mg ha$^{-1}$ would be required to supply the N needs of the crop. This amounts to the addition of 0.88, 0.94, 1.8, 11, and 26% for P2, P3, P4, P5, and P6 biochar, respectively, to the soil (assumed ~2000 Mg ha$^{-1}$). The addition rates for the N supply with P5 and P6 are unreasonable for agricultural production systems. However, the low temperature pyrolysis PL ($\leq$400 °C) could contribute a great amount of available N, especially ammonium nitrogen, indicating the potential to act as a soil N amendment under a reasonable addition rate.

*4.2. Effects on Carbon Mineralization*

Soil C mineralization was significantly affected by the addition rate and pyrolysis temperature of the PLB (Figure 2, Table S2). Without additional organic material input, PLBs were the only additional C source in studied soil. The increase in the $CO_2$–C release could be attributed to the minerals or compounds that are easily released from the studied PLBs, especially for the lower temperature-pyrolyzed PLBs, such as P2 and P3, which have a similar organic fertilizer application as soil, and lead to a significant ($p < 0.05$) effect on C losses like $CO_2$ [23]. In this study, the low temperature pyrolysis PLB (P2 and P3) application produced more water-extractable C. After adding biochar, the soil respiration increased. The result is that biochar is not as inert as the public believes but rather provides a large amount of unstable C, which is used as an energy source for soil microorganisms [42]. An unstable biochar C part may be more easily degraded by microorganisms than other biochar parts, so it is easy to obtain soon after adding biochar to soil microorganisms. The higher WEOC in P2, P3, and P4 biochar indicates that a more effective C can be applied after adding these biochars and has the potential to enhance C mineralization (increasing soil $CO_2$ release). On the contrary, the P5 and P6 biochar have less potential to enhance C mineralization due to the smaller scale of WEOC. In addition to the presence of more unstable C fractions in the substrates, the higher C mineralization rate may also include a priming-effect component, which is caused by the increase in the N input caused by the application of the matrix to the native soil C [38]. In all cases (biochars), the percentage of available N in the form of nitrate is <0.01% of the total [10]. Biochars frequently have low concentrations of extractable mineral N (as $NO_3^-$ and $NH_4^+$) [43] and this is due to the loss of gaseous N during pyrolysis [44]. In fact, P2, P3, and P4 biochar show a high availability of N (11.4, 10.6, and 5.65 g kg$^{-1}$, respectively), relative to P5 and P6 (0.94 and 0.39 g kg$^{-1}$) (Table 1) and this could be responsible for the inducing of the aforementioned priming effect. Moreover, in this study, the amount of available N as nitrate ($NO_3^-$) is negligible for all the PLBs, but the cumulative content of the total water-extractable inorganic N ($NH_4^+$–N) of P2, P3, P4, P5, and P6, as compared to total N, was about 8.4, 7.0, 2.2, 0.12, and 0.10%, respectively. The results of the cumulative $NH_4^+$–N of the five studied PLBs were much higher than the reports concerning lignocellulosic biochar [19,21]. In addition to the enhancement of more available N in P2, P3, and P4 biochar, the higher water-extractable N could also enhance the C mineralization rate.

Aromatic C is a recalcitrant ingredient of biochar and is mainly responsible for the stability of biochar in the soil; however, biochar also contains relatively unstable components, including aliphatic C, carboxyl groups and carbohydrates, which are easily mineralized [45]. As an indicator of the presence of polar functional groups, the O/C ratio can provide information on the surface hydrophilicity and hydrophobicity of the charred material [46]. In addition, the value of the H/C ratio can provide information on dehydrogenation resulting

from the extensive carbonization produced by thermal induction [47]. Because of the dehydration of volatile organics [48], H, N, and O contents were found to decrease with the increase in production temperature [49]. Correspondingly, the O/C (polarity), (O+N)/C (polarity), and H/C (aromaticity) values decreased with the increase in the pyrolysis temperature because of dehydration and decarboxylation reactions [11]. Windeatt et al. [50] showed that, as the pyrolysis temperature increases, the aromaticity (H/C) increases and the polarity (O/C) decreases significantly. At a higher temperature, with a lower O/C ratio, an aromatic ring structure is formed and depicts a stable crystal, that is, a graphite-like structure [11]. A low O/C ratio indicates a relatively high degree of aromaticity and a reduced hydrophilicity [51] and could be attributed to the greater extent of carbonization that is caused by the removal of H– and O– containing functional groups from the original feedstock [52]. The O/C ratio is in the sequence of P2 (0.76) > P3 (0.67) >P4 (0.56) > P5 (0.25) = P6 (0.25) (Table 1). According to the report by Pariyar et al. [11], the half-lives of P2 and P3 are <100 years because the O/C ratio is >0.6; in addition, the half-lives of P4, P5 and P6 are between 100 and 1000 years because the O/C ratio is between 0.2 and 0.6. In addition, the H/C ratios of P2, P3 and P4 are much higher than 0.7, indicating a lack of condensed aromatic structures and a possible loss of its overall stability [46], which may explain the higher C mineralization in these PLB treatments. The H/C ratio of P5 and P6 is close to 0.7, indicating that the substance has some highly condensed aromatic ring systems. Schimmelpfennig et al. [46] recommended that biochar, with H/C $\leq$ 0.6 and O/C $\leq$ 0.4, has stable characteristics and can be used in soil applications to sequester C. In this study, P5 and P6 biochars may have the potential for C sequestration, as evidenced by the longer half-life and MRT in the amended soil C mineralization (Table 2).

The analysis of the applied C mineralized (ACM) during the incubation period is helpful and is a good tool to ascertain when the new equilibrium in the soil microbial population is reached [53]; in other words, it is used to assess whether the soil microbial population reached the steady-state equilibrium [23]. As shown in Figure 3, P5 and P6 treatments reached steady-state equilibrium quickly, indicating a relatively lower soil microbial population. Furthermore, due to the much smaller C pools (labile C + recalcitrant C) (<10%) in P5 and P6 treatments, the half-life and MRT of ACM are shorter than P2, P3, and P4 treatments (Table 3). However, it is worth noting that the recalcitrant C pools of P5 and P6 treatments has only one-fifth to one-tenth of the half-life and MRT are not very short, as compared to P2, P3, and P4 treatments. Composts with cumulative ACM of about 2.5% can be considered to possess sufficiently stabilized organic matter [54]. In this study, all PLB treatments almost had a higher cumulative ACM than 2.5%, indicating that these PLBs may be used as organic fertilizers (like compost). In addition, the application of stable and mature compost to the soil resulted in $CO_2$–C emissions < 250 mg C (applied g organic C)$^{-1}$ (25%) [55]. In the current work, the 1%, 5%, and 10% applications of P2 and P3 led to higher values. Compared to the lower temperature PLBs (<400 °C), the OM in the PLB with a pyrolysis > 400 °C was more stable, highlighting a higher resistance to mineralization. Therefore, PLB with a pyrolysis > 400 °C can be regarded as a mature substrate.

## 5. Conclusions

The current research results show that the pyrolysis temperature is a key factor in determining the quality of PL slow pyrolysis biochar, and it plays an important role in C sequestration in PLB-amended soil. The yield, available N and CEC value of PL biochar, as well as the WEOC and $NH_4^+$–N concentration decreases with the increase in the pyrolysis temperature in the range of 200–600 °C, while the C stability (low O/C and H/C values), pH, EC, available K and Ca, total P, K, Ca and Mg increase. The soil C mineralization and ACM results both suggested that the PLB pyrolyzed at more than 400 °C can be considered to be a mature substrate and have a higher potential to sequestrate C in acidic soil. However, due to the potential risk of soil salinization, the application rate should not be higher than 5%. In the present study, to produce both an agricultural-use and a

C sequestration-use PL biochar, a slow pyrolysis temperature within 400–600 °C or PLB with an O/C atomic ratio < 0.6 and 0.7 < H/C atomic ratio < 1.5 should be adopted. The appropriate addition rate of such a PLB would be between 1% to 5% in acidic soil in order to improve the soil nutrient conditions, to sequestrate soil C, and to reduce soil salinization. As a sustainable management approach, further investigations should be conducted on short-term and long-term field applications in order to test the results.

**Supplementary Materials:** The following are available online at https://www.mdpi.com/article/10.3390/agronomy11091692/s1, Table S1: Cumulative C mineralized during the incubation period; Table S2: Significant difference of $CO_2$–C release of nine times monitoring between addition rates and treats compared to the control during 56 days of incubation; Table S3: Significant test of cumulative $CO_2$–C, and water extraction solution pH and electrical conductivity after 56 days incubation.

**Author Contributions:** Conceptualization, C.-C.T. and Y.-F.C.; methodology, C.-C.T. and Y.-F.C.; validation, C.-C.T. and Y.-F.C., formal analysis, Y.-F.C.; investigation, C.-C.T. and Y.-F.C.; data curation, C.-C.T. and Y.-F.C.; writing-original draft preparation, C.-C.T.; writing-review and editing, C.-C.T.; supervision, C.-C.T.; funding acquisition, C.-C.T. All authors have read and agreed to the published version of the manuscript.

**Funding:** This research was funded by Ministry of Science and Technology of the Republic of China, Contract number MOST 105-2313-B-197-001-.

**Data Availability Statement:** Data will be made available upon request.

**Acknowledgments:** The authors would like to thank the staffs of Elemental Analysis, the Instrumentation Center, National Taiwan University, for help with C, H, N, O, and S contents. Thank Su-Yun Fang (Instrumentation Center at National Tsing Hua University (NTHU)) for Bruker AVIII-400 MHz Solid NMR analysis (MOST 110-2731-M-007-001-). Specially thank Keng-Tung Wu, (Biomass Energy Research Lab., Department of Forestry, National Chung Hsing University, Taichung, Taiwan) for poultry litter biochar production.

**Conflicts of Interest:** The authors declare no conflict of interest.

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
