# Peer review of "Quality Evaluation of Poultry Litter Biochar Produced at Different Pyrolysis Temperatures as a Sustainable Management Approach and Its Impact on Soil Carbon Mineralization"

_agronomy, doi:10.3390/agronomy11091692_

Round 1
Reviewer 1 Report
Excellent job on this paper. It is well presented, scientifically sound, and subject matter is important. However, minor revisions is needed on some of the sentences, which needs to be rewritten. See comments on the manuscript.

Author Response
Dear reviewer:
Thank referee’s valuable comments. The original manuscript has undergone English language editing by MDPI (English Editing ID: english-33403). We have revised the manuscript carefully and in details based on the valuable comments of reviewers, and have made the presentation and discussion of manuscript more complete. Please see the attachment.
best regards

Reviewer 2 Report
This manuscript investigated the effects of pyrolysis temperature of poultry litter biochar on rate on soil properties.
However, this manuscript exist some issue as following.
- The novelty of this study is not high enough. While current studies on temperature effect of biochar on C mineralization have been well studied by both lab study and field study, current study design is too simple with only soil respiration experiment, not enough for soil C mineralization study which typically include soil microbial control and added C substrates treatment to support the well discussion of the soil C mineralization process.
- In introduction, the authors listed the inconsistent findings of biochar with different feedstocks as well as different soil types based on previous literature, however, in result and discussion, the comparison on studied poultry litter biochar and acid soil with other studies lack further discussion.
- This manuscript exist grammar issues that cause misleading (eg. Line 52-55, 59-62, 298-300...). Also, the use of abbreviation in the beginning of a sentence should be avoid (eg. Line 80: “PL slow pyrolysis at 300oC....”).
Author Response

(The authors gave the same response as above.)

Reviewer 3 Report
The current article evaluating the quality of poultry litter biochar is interesting. However, it needs some minor corrections to improve the article.
1.) Numerical values related to results must be included in the abstract.
2.) Please include some relevant points of aromaticity and aliphaticity in the Introduction section.
3.) The characteristics studies showed in Table 1, e.g., Aromatic C, Phenolic C, etc. have to be briefed in the Materials and Methods section. A detail of 13C Nuclear Magnetic Resonance (NMR) spectra is required including temperature, wavelength, angle, etc.
4.) Provide the formula of CO2-C release and cumulative C release in the Materials and Methods section.
5.) Line 189…..The Pearson correlation coefficient (r) was also calculated…….Where this has been presented in the article??
6.) Line 426, 427…..Recheck the symbols (~) used, whether it is approximately or range
7.) Line 431…..% should be added in the last….e.g., 0.88, 0.94, 1.8, 11, and 26%......Similarly, in Line 464…….Maintain uniformity as presented in Line 459…..11.4, 10.6, and 5.65 g kg-1.
8.) Line 458….. NO3, NH4 and NO2 …..The charges are not included. Give comma after NH4+.
9.) Improve the discussion…describe the aromatic or long-chain hydrocarbons in detail wherever applicable.
10.) Check Line 517….. “produce both agricultural- and C sequestration-use…” Rephrase.
11.) Rephrase…… “The amendment rate could higher than 1%”
12.) The present article is very shortly presented. Please describe each part in detail so that the readers do not get confused. For example, the half-life (t1/2), mean residence time (MRT), labile C pool, etc. must be elaborated in the context of the article rather than only presenting data or writing results.
13.) Check all the typos and grammatical mistakes throughout the manuscript.
Author Response

(The authors gave the same response as above.)

Round 2
Reviewer 2 Report
Manuscript has been well revised, well organized with clear logic and sufficient evidence. Acceptance is recommended.